# Xanthogranulomatous Pyelonephritis Caused by *Stenotrophomonas maltophilia*—The First Case Report and Brief Review

**DOI:** 10.3390/pathogens11010081

**Published:** 2022-01-10

**Authors:** Răzvan-Cosmin Petca, Răzvan-Alexandru Dănău, Răzvan-Ionuț Popescu, Daniel Damian, Cristian Mareș, Aida Petca, Viorel Jinga

**Affiliations:** 1“Carol Davila” University of Medicine and Pharmacy, 050474 Bucharest, Romania; razvan.petca@umfcd.ro (R.-C.P.); razvan.danau@umfcd.ro (R.-A.D.); cristian.mares@drd.umfcd.ro (C.M.); aida.petca@umfcd.ro (A.P.); viorel.jinga@umfcd.ro (V.J.); 2Department of Urology, “Prof. Dr. Theodor Burghele” Clinical Hospital, 050659 Bucharest, Romania; daniel_damian2004@yahoo.com; 3Department of Obstetrics and Gynecology, Elias University Hospital, 011461 Bucharest, Romania

**Keywords:** *Stenotrophomonas maltophilia*, Xanthogranulomatous pyelonephritis, XGP, urinary tract infection, renal stone, nephrectomy

## Abstract

Xanthogranulomatous pyelonephritis (XGP) represents a rare and severe pathology secondary to chronic urinary obstruction and recurrent infections. Commonly, this condition leads to loss of kidney function, and frequently, surgical approach is the only optional treatment. *Proteus mirabilis* and *Escherichia coli* are the most frequent pathogens associated with XGP. The actual changes in the pathogen’s characteristics increased the risk of newly acquired infections once considered opportunistic. *Stenotrophomonas malthophilia* is one of those agents more related to immunocompromised patients, presenting an increased incidence and modified antibiotic resistance profile in the modern era. This case report presents a healthy female patient with an underlying renal stone pathology diagnosed with XGP related to *S. maltophilia* urinary infection. After a complete biological and imagistic evaluation, the case was treated as pyonephrosis. Empirical antibiotic administration and a surgical approach were considered. A total nephrectomy was performed, but the patient’s condition did not improve. The patient’s status improved when specific antibiotics were administered based on the bacterial identification and antibiotic susceptibility pattern of drained perinephric fluid. Levofloxacin and Vancomycin were considered the optimal combination in this case. The histopathological examination revealed XGP secondary to chronic renal stone. The present study describes the first case of XGP related to an aerobic Gram-negative pathogen such as *S. maltophilia*, once considered opportunistic, in an apparently healthy female adult.

## 1. Introduction

According to the World Health Organization’s newest warnings, increasing antibiotic resistance is already defined as one of the major concerns of global health status [1]. Besides the common pathogens, many unusual bacteria, once considered opportunistic, seem to spread even in normal healthy adults and become another important aspect as they are hard to identify and difficult to treat.

*Stenotrophomonas maltophilia* is an aerobic Gram-negative bacterium commonly described as an opportunistic and nosocomial pathogen. Hospitalized and immunocompromised patients are more likely to be affected, as this bacterium presents high adherence for the respiratory, gastrointestinal, and urinary systems [2].

*S. maltophilia* infections present high morbidity and mortality rates for specific reasons. Antimicrobial resistance increases when antibiotics are administered for extended periods, patients’ poor health status complicates drug administration, and the species has a unique characteristic of high resistance [3,4,5].

This opportunistic bacterium has been reported to be associated with a wide variety of infections most commonly related to the respiratory tract, such as pneumonia and acute exacerbations of chronic obstructive pulmonary disease [6,7].

Xanthogranulomatous pyelonephritis (XGP) is an aggressive form of chronic infection usually associated with lithiasis and impaired renal drainage, which can be easily misdiagnosed with renal cancer because of unspecific imagistic characteristics [8]. This disease was explicated by Schlagenhaufer for the first time in 1916 [9]. Though common in the fifth to sixth decade, it can occur at any age, and women are more frequently affected than men [10]. Commonly, it presents a unilateral distribution and is often characterized by the loss of the renal unit [11]. The general opinion agrees that *Escherichia coli* and *Proteus* are the most frequent pathogens related to XGP [12,13,14,15,16].

Although a variety of infections associated with *S. maltophilia* have been reported, this paper aims to describe the first case of XGP associated with a *Stenotrophomonas maltophilia* isolate.

## 2. Case Report

A 41-year-old female was admitted to Prof. Dr. Th. Burghele Clinical Hospital, one of the high-volume urology centers in Bucharest for diffuse abdominal pain, which was more pronounced in the left flank and altered general condition. Inspiratory dyspnea and fatigue completed the presentation symptoms that progressively developed in the last two weeks.

Past untreated nephroureterolithiasis and normochromic normocytic anemia were registered in the medical records. A few months before, the patient had undergone a double J stent placement in an emergency hospital to assure normal renal drainage as she presented high-grade uretero-hydronephrosis. The definitive surgical treatment was delayed once the hospital where she was first admitted became an exclusive COVID-19 support hospital as a governmental emergency decision in the 2020 pandemic situation.

The physical examination revealed abdominal pain and tenderness associated with a palpable mass in the lumbar area. The body temperature was evaluated within the normal range at admission. The cardiovascular and respiratory evaluation revealed signs of tachycardia (100 beats/min.) and tachypnea (>30 respiratory rate/min.). Measured blood pressure was 100/80 mmHg, and oxygen saturation was 95%.

SARS-CoV-2 infection was excluded after rapid antigen testing, and a PCR probe was also collected, and both were negative. The patient also declared no previous symptoms attributable to COVID-19 infection.

The patient was assessed with complete blood tests, which revealed leukocytosis (WBC: 17.7 × 10^9^/L), anemia (Hgb: 8.7 g/dL), and thrombocytosis (PLT: 629,000/μL). Increased values were obtained for CRP (311 mg/dL) and D-dimers (1,238 ng/mL). The procalcitonin level was within the normal range. The urine culture, first performed when the patient was primarily assessed, revealed no signs of infection (<10^3^ CFU).

A CT was performed after correlating the clinical and laboratory findings to examine the thorax, abdomen, and pelvic regions. The imagistic report included a left renal inflammatory process more likely developed on untreated staghorn lithiasis. Important adhesion processes occurred between the kidney and the surrounding structures and altered the normal anatomy of the posterior peritoneum, psoas muscle, and left diaphragm. The relation between the kidney and the surrounding organs and the lithiasis pathology is reflected in Figure 1 and Figure 2.

Severe left pyonephrosis was suspected. A broad-spectrum antibiotic therapy was started, including Meropenem (1g at 8 h) + Metronidazole (500 mg at 6 h) + Amikacin (500 mg at 12 h) for two days before surgery, along with prophylactic anticoagulation—Enoxaparin, estimated according to the patient’s body mass index (40 mg). Hemodynamic and hydro-electrolytic rebalancing was also managed.

Exploratory lumbotomy with left nephrectomy—Figure 3—was performed under general anesthesia. Approximately 200 mL of purulent fetid fluid was extracted after reaching the retroperitoneal space. The perirenal area was hardened, with adhesions to the psoas and diaphragm muscle and the posterior peritoneum, where the macroscopic structure was altered. Two drainage catheters were left in the lumbar area. Part of the posterior peritoneum and the psoas fascia was excised because of the highly inflammatory developed process. Despite the increased level of difficulty, no major complications were registered during surgical intervention.

Until the bacteriological exam of the drained fluid highlighted the presence of *Stenotrophomonas maltophilia*, the evolution with broad-spectrum antibiotics did not provide satisfactory results, and the blood tests continuously revealed leukocytosis (day 1—WBC: 16.3 × 10^9^/L, Hb: 8.7 g/dL; day 3—WBC: 14.0 × 10^9^/L, Hb: 8.1 g/dL; day 5: WBC: 12.4 × 10^9^/L; day 6—WBC: 12.0 × 10^9^/L, Hb: 9.4g/dL; day 8—WBC: 9.4 × 10^9^/L, Hb: 10.8 g/dL). Clinically, persistent fever that ranged within 38 and 39.6 Celsius degrees also represented an alarming sign.

The antibiogram for the bacterial testing indicated susceptibility for *S. maltophilia* to colistin, trimethoprim/sulfamethoxazole, aminoglycosides, fluoroquinolones, and tetracycline. Hence, a combination of Levofloxacin and Vancomycin started to be administered and significantly impacted the patient’s condition. The clinical symptoms, including persisting fever, fatigue, and altered general state, were remitted within three days, and the favorable evolution continued at 14 days after hospital discharge during the active surveillance.

The anatomopathological result was XGP. The complete description revealed a lesion at the superior pole, 11 × 7 × 6 cm, with a soft, mottled, tan, focally cystic cut surface. It was well delimited from the adjacent renal parenchyma and expanded, but did not infiltrate, the overlying capsule from which it is readily dissected.

On microscopy, representative sections examined showed granulomatous mixed inflammatory cell infiltrate. Sheets/Aggregates/Nodules of xanthomatous histiocytes with an admixture of neutrophils, lymphocytes, and plasma cells in various numbers were seen—Figure 4. Multinucleated histiocytic giant cells were noteworthy. Areas of fibrosis were also observed.

Standard microbiological techniques (microscopy, culture characteristics, and oxidase reaction) were used for the species identification of *Stenotrophomonas maltophilia*, utilizing the API-20NE system (bioMérieux, Marcy l’Etoile, France). We tested the antimicrobial susceptibility for *S. maltophilia* isolate with the agar dilution method. The microbiological laboratory of the hospital preserved the isolate frozen at −80 °C. The antibiotics tested included ceftazidime, ticarcillin/clavulanic acid, amikacin, gentamicin, chloramphenicol, ciprofloxacin, tetracycline, colistin, and trimethoprim/sulfamethoxazole. The results were interpreted according to the Clinical and Laboratory Standards Institute (CLSI) criteria [17]. ATCC 25922 *Escherichia coli* and ATCC 27853 *Pseudomonas aeruginosa* were utilized as quality control strains.

## 3. Discussion

*Stenotrophomonas maltophilia* is one of the rising new pathogens usually associated with nosocomial infections, which mainly affects immunosuppressed and debilitated patients. The most frequent source of bacteriemia for this opportunistic agent is represented by the infected respiratory tract, gastrointestinal system, or a central venous catheter [18]. Rarely, the urinary tract appears to be responsible as the primary source. A recent study conducted by Gajdács et al. in Hungary that analyzed a 10 year period, including more than 9000 urine positive cultures, collected from both outpatients and inpatients, reported only 11 cases of *S. maltophilia* [19]. It is supplementary proof to attest to the rarity of our case.

To the best of our knowledge, this is the first recorded case of XGP caused by this opportunistic bacterium in an apparently normal healthy female patient without any diagnosed immunosuppressing diseases.

XGP is a severe retroperitoneal disease with an incompletely elucidated etiology and appears as a result of long-term obstructive uropathy and recurrent infections [16,20,21]. The certain diagnostic in this pathology can only be set after histopathological examination because it presents a lack of clinical and imagistic specific characteristics and can be easily misdiagnosed with renal carcinoma [22].

Chronic inflammation is generally limited to the renal parenchyma, and severe damage consists of the complete loss of kidney function. Still, the literature findings also report the involvement of the surrounding organs [23]. Based on clinical and pathological aspects, Malek and Elder staged XGP into three phases according to natural disease evolution [24]. The first stage is defined as only involving the renal parenchyma. In the second stage, the infectious process spreads into the renal fat, and in the third stage, it extends to the surrounding structures of the retroperitoneum [24]. According to this classification, this case was staged as a third phase since the inflammation process affected the posterior peritoneum, the psoas muscle, and the diaphragm.

The presenting clinical symptoms are unspecific and include abdominal pain, fever, weight loss, hematuria, dysuria, and palpable mass [12]. Our patient complained about diffuse abdominal pain, altered general condition, inspiratory dyspnea, and fatigue.

Imagistic evaluation is poorly represented as ultrasound and kidney, ureter, and bladder X-rays cannot provide conclusive details. An abdominal CT scan is considered the option of choice for these cases and often reveals renal stones, a dilated pyelo-calyceal system, and low-density mass lesions [25]. In this case, the CT evaluation suspected an inflammatory process developed on untreated renal stone disease and suggested adhesion presence between the kidney and the surrounding organs.

Infection represents one of the main conditions in developing XGP. The most commonly reported pathogens isolated in urine samples were *Proteus mirabilis* and *Escherichia coli* [14,15]. Although the urine tests performed at admission revealed no signs of infection, the bacteriological samples of the drained retroperitoneal area revealed the presence of *Stenotrophomonas maltophilia*. Even though broad-spectrum antibiotic therapy was administered, including Meropenem, Metronidazole, and Amikacin three days before surgery, the patient’s condition displayed no significant favorable changes after nephrectomy. After receiving the culture and antibiogram results, switching the medical treatment to Levofloxacin and Vancomycin improved the evolution course. After three days, the symptoms completely remitted, and blood tests regained their values within the normal range.

*Stenotrophomonas malthophilia* is a Gram-negative bacterial pathogen that was isolated for the first time in 1963 as *Pseudomonas maltophilia* [26]. It is considered a waterborne germ, and several related infection sources except natural spread were described, including contaminated endoscopes [27]. In this case, it was recorded that the patient had two previous endoscopic interventions for renal stone with ureteral stent placement in some other medical units. Even though it is not considered a highly virulent bacteria, it became an important pathogen for hospital-acquired infections and can have a significant impact on immunocompromised patients. Most commonly, it was reported as a causative agent for respiratory tract infections [28]. It can also affect other organs such as bones, the biliary duct, eyes, and the urinary tract. It was also found to be involved in the development of endocarditis and meningitis [29,30]. An unusual aspect of this case was that the patient had no underlying pathologies to suggest an immunocompromised status as she was tested for HIV, Hepatitis B, and C, and no evidence of cancer disease was found after the complete clinical evaluation. Some underlying conditions such as neutropenia and structural abnormalities mainly represented by urinary tract obstruction were identified by Vartivarian et al. in their study after analyzing 15 cases of complicated and uncomplicated urinary tract infections caused by *S. maltophilia* [31]. The only associated condition found was an incomplete-treated renal stone disease. An essential aspect of the case was the presence of a left double J stent and sterile urine culture, which did not help for preoperative antibiotic administration.

Because *S. maltophilia* is usually a non-pathogenic organism or colonizer, it is generally not initially suspected as a causative bacterium [31]. Distinguishing between colonization and infection due to this organism is complex. In this case, a diagnosis of XPG due to *S. maltophilia* was not so easy. The management of associated infection may be problematic because of its high intrinsic or acquired resistance to multiple broad-spectrum antibiotics due to various mechanisms such as decreased permeability, the production of β-lactamases, and aminoglycoside modifying enzymes the presence of multidrug efflux pumps [32].

Some studies indicate Trimethoprim/Sulfamethoxazole (TMP/SMX) as the treatment of choice according to the good results obtained in vitro experiments on these isolates [33]. Recent data reveal 84% susceptibility for TMP/SMX after analyzing *S. malthophilia* samples over a 10 year period [34]. This demonstrates 16% of patients to be resistant to the once considered first-line treatment. Additionally, in the same study, high levels of resistance to Levofloxacin and colistin were also identified [34].

In vitro testing acknowledged Fluoroquinolones as an important treatment option for this kind of infection. Recent studies revealed increased resistance rates worldwide for Levofloxacin, which severely impacted therapeutic alternatives [35]. In this case, Levofloxacin was successfully used in combination with Vancomycin according to laboratory antibiogram data. The isolates also showed relatively high susceptibility to colistin, trimethoprim/sulfamethoxazole, aminoglycosides, fluoroquinolones, and tetracycline.

A retrospective study compared TMP/SMX with Minocycline results obtained after treating 45 patients with diagnosed infection. The reported treatment failure was 41% for TMP/SMX and 30% for Minocycline, but no differences were recorded in the overall mortality rates [36].

Cefidecorol, a novel siderophore cephalosporin, was successfully tested against *S. maltophilia* [37].

XGP is a rare condition, and a precise diagnosis can only be established based on histopathological examination, which consists of highlighting the presence of lipid-laden foamy macrophages [20]. The indication treatment is represented by a combination of broad-spectrum antibiotic administration and a surgical approach. Depending on the disease stage, the surgical intervention can be partial or total nephrectomy [38]. In this case, total nephrectomy and partial excision of the posterior peritoneum and psoas fascia were necessary to remove the infectious process altogether.

According to the rarity of this disease, there is no consensus or guidelines for an optimal antibiotic treatment or other data related to the drug administration period or the optimal surgical moment.

Some literature findings reveal that XGP can be successfully treated with antibiotics alone in cases of a focal disease, which involves less than 10% of the renal parenchyma [39]. Serial ultrasonography evaluation is necessary when choosing the main approach line of medical treatment.

## 4. Conclusions

To the best of our knowledge, this is the first reported case of XGP caused by *S. maltophilia* in a young female patient without any immunocompromising diseases.

The absence of signs that could have predicted a complicated evolution after the surgical procedure is especially notable. This case’s key point was broad-spectrum antibiotic administration ab initio and immediate drug adjustment after microbiological examination of the extracted fluid. We conclude that treating Xanthogranulomatous pyelonephritis implies bacterial strain identification and prompt antibiotic treatment, correlated with a proper surgical approach.

As *Strenotrophomonas maltophilia* becomes an emerging infectious agent, it is mandatory to investigate and monitor its main characteristics, spread, persistence, and antibiotic resistance profile further.

## Figures and Tables

**Figure 1 pathogens-11-00081-f001:**
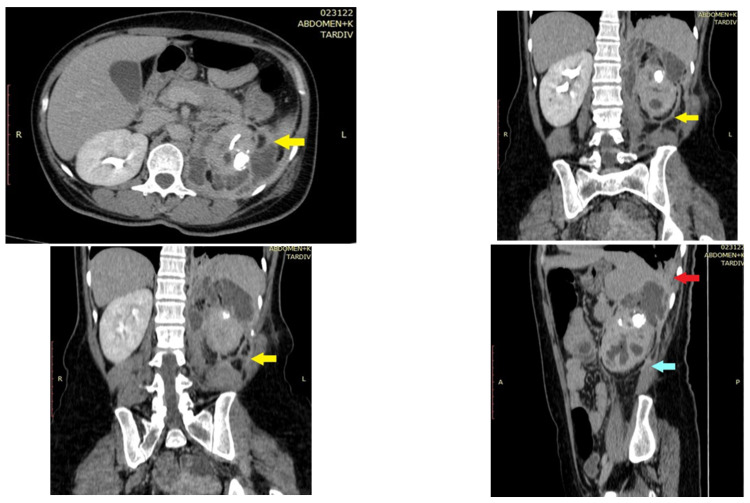
CT sections (axial, coronal, and sagittal) reveal the infectious process extension and relation with surrounding organs. The intimate contact of the inflammatory process with the diaphragm muscle at the upper pole (red arrow), the changed aspect of the peri nephritic fat (yellow arrow), and the relation with the psoas muscle can be observed (blue arrow). The posterior peritoneum seems to also be involved with traction close to the kidney.

**Figure 2 pathogens-11-00081-f002:**
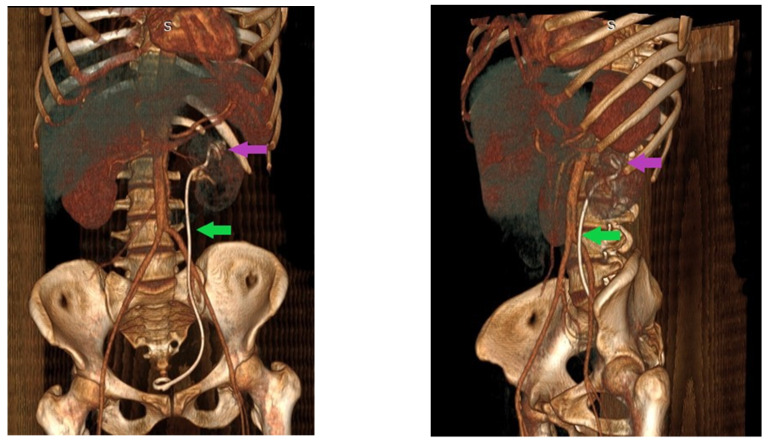
CT 3D reconstruction on arterial time revealing the double J stent position (green arrow), the presence of renal left renal stones (purple arrow), and kidney position.

**Figure 3 pathogens-11-00081-f003:**
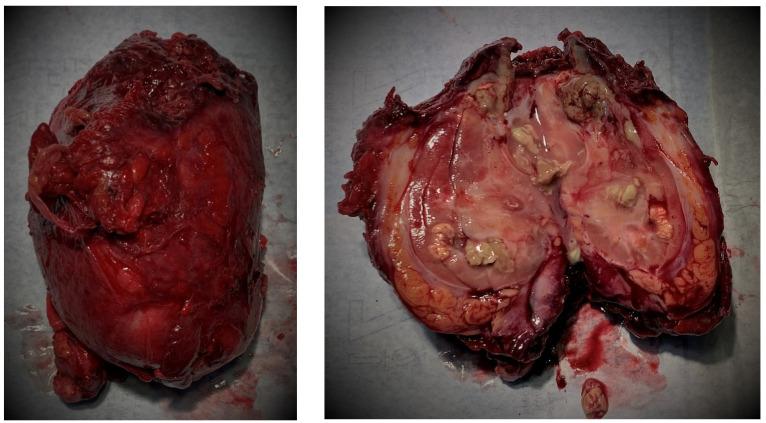
Post nephrectomy images before and after sectioning the left kidney. In the left image, it can be observed the macroscopical aspect characteristic of Xanthogranulomatous Pyelonephritis.

**Figure 4 pathogens-11-00081-f004:**
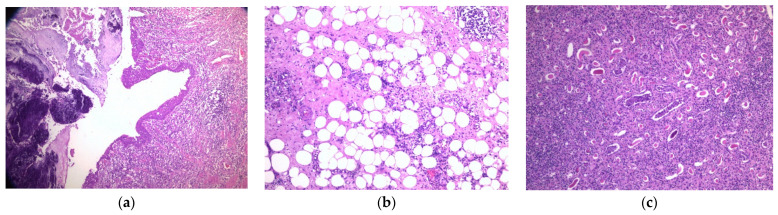
Anatomopathological details. Hematoxylin–eosin stain (HE). (**a**) Inflammatory infiltrates extended in the pyelocaliceal area and ureter. Area of ulceration, granulation tissue, and reactive epithelial modifications. Multiple polymorphonuclear neutrophiles with intraepithelial extension and microabscesses. HE-10x; (**b**) extension of the inflammatory process in the perirenal adipose tissue. HE-10x; (**c**) renal tubules with pseudo thyroid phenomena. In the center, intratubular microabscesses. HE-20x.

## Data Availability

Data sharing not applicable.

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
