# Peer review of "Xanthogranulomatous Pyelonephritis Caused by Stenotrophomonas maltophilia—The First Case Report and Brief Review"

_pathogens, 2022, doi:10.3390/pathogens11010081_

Round 1

Reviewer 1 Report

This is an interesting case of XGP with S maltophilia. The readers need some idea as to why the 41- year old woman have a staghorn calculus (?) with obstructed ureter that needed a double J stent. That is not normal. Did she have any chronic underlying infections or conditions that led to it. Was she taking any medications that could have suppressed her immune system. Did she have a work up/imaging study or any cultures at the other hospital where she received the stent? What was the result of the COVID PCR test? Had she contracted COVID previously? Many of these questions need to be answered.

The manuscript also needs to be reviewed for language.  

Reviewer 2 Report

This is an interesting report about a case of xanthogranulomatous pyelonephritis due to Stenotrophomonas Malthophilia. The radiological and pathological images shown are very representative. Y have some minor comments.

-Abstract: line 17: “Proteus mirabilis and Escherichia Coli are mostly associated with XGP”. The meaning of this sentence is not easy to understand the way it is written. P Mirab and E.Coli are the most frequent pathogens? In fact, this same sentence is correctly described in line 59.

-Line 39 and line 43: The use of “ as long as” meaning “providing that” makes no sense.

-Line 74 What does “plain pandemic situation” stand for?

-Figure 1: I suggest using colour arrows to point to the characteristic findings described : intimate contact of the inflammatory process with the diaphragm muscle at the upper pole, the changed aspect of the peri nephritic fat, and the relation with the psoas muscle.

-The same for figure 2

-Line 134-135: please locate “Figure 4” correctly

-Line 185: what does KUB stand for? Again “ as long as” is probably not correct here

Reviewer 3 Report

Major comments:

The report is the first case of XGP related to an aerobic gram-negative pathogen like S. maltophilia.

Minor comments:

The paper below should be cited.

Vartivarian SE, Papadakis KA, Anaissie EJ. Stenotrophomonas (Xanthomonas) maltophilia urinary tract infection. A disease that is usually severe and complicated. Arch Intern Med. 1996 Feb 26;156(4):433-5.
